# Does Skill Polarization Affect Wage Polarization? U.S. Evidence 2009–2021

**Huajie Jiang** [ID] **and Qiguo Gong** *

School of Economics and Management, University of Chinese Academy of Sciences, 80 Zhongguancun East Road, Haidian District, Beijing 100190, China
* Correspondence: gongqg@ucas.ac.cn

**Abstract:** (1) Background: Wage polarization and skill polarization are frequently mentioned in the literature, but relatively few empirical studies have focused on the relationship between skill polarization and wage polarization. (2) Methods: Using occupation–skill data from the O*NET database in the United States from 2009 to 2021, this study constructs the occupational socio-cognitive skill scores and the number of perceived physical skills effectively used by an occupation as proxies for measuring skill polarization and matches the Occupational Employment and Wage Statistics data from the corresponding years to explore the relationship between skill polarization and wage polarization by using 2SLS. (3) Results: Increases in both the occupational socio-cognitive skills scores and the number of sensory–physical skills effectively used by an occupation lead to higher wages, but the magnitude of the positive effects of these two indicators are different. We also find that these control variables can reduce occupational wages with a lagged effect. (4) Conclusion: Our findings confirm that skills polarization has a positive effect on wage polarization, providing new insights into understanding employment inequality in the labor market. Authorities should focus more attention on increasing the earnings of the low- and middle-skilled workers, especially through vocational skills training to increase the number of sensory–physical skills that can ultimately mitigate wage polarization.

**Keywords:** skill polarization; wage polarization; panel data; 2SLS

## 1. Introduction

In recent decades, many labor markets around the world have shown a "hollowing out of the middle class", that is, middle-class jobs are disappearing and wage inequality is increasing. Job polarization often coincides with wage polarization [1]. In some studies, occupational income and employment shares are often used as the data source for analyzing employment polarization [2]. Since the groundbreaking research of the job polarization phenomenon by Autor et al. (2003) [3], several papers have addressed job polarization and wage polarization [4–9]. Existing studies are divided on whether wage changes in the labor market are signs of inequality or wage polarization, but this difference is mainly due to the different perspectives on measuring wage changes [1,7,10–17].

Studies found that, in the labor market of the U.S., Italy and other countries, wages and employment shares at the tails of the skill distribution (i.e., high-skilled and low-skilled workers) steadily increased, while workers in the middle faced a crisis of stagnant wages and declining employment share [2,13,18–21]. The existing literature reveals that technological strategies [22,23], international trade [24], foreign capital inflows [25], location [26] and other factors can affect wage polarization. This study refers to this change in labor market wages, which is similar to job polarization, as wage polarization.

The polarization of skills in the labor market is also of concern. Studies have found that the upgrading of skills in the European labor market is accompanied by polarization [27]. Skills in the U.K. labor market became polarized in the early 1980s [28]. Some scholars agree

that the U.K. labor market has experienced skill polarization, but question the existing data collection and measurement methods for studying this phenomenon [29]. Skill polarization occurs primarily in labor markets characterized by high levels of day-to-day activities and human capital, where middle-skilled jobs are largely replaced by advances in information technology [2]. Employers are changing in terms of skill requirements; soft skills are replacing technical skills and the adoption of new skills results in skills polarization [30]. Alabdulkareem et al. (2018) [31] and Xu et al. (2021) [32] pioneered the study of proxies for skill polarization starting from occupation–skill data and examined the phenomenon of employment polarization in the U.S. and China, respectively.

Technological progress [33], skills agglomeration [34], trade [35] and other factors may cause skill polarization. Some studies revealed the phenomenon of skill polarization in the labor market while focusing on job polarization [2,27,30]. It should be pointed out that, although there have been many studies on the issue of skill polarization, including many attempts to use data to conduct empirical research, the existing literature has not yet reached a consensus on what indicators to use to accurately describe the issue of skill polarization. This is primarily related to the strategies used to measure skills in existing studies. For example, years of education are often used as a proxy for skill level, so some studies have used changes in the employment share of the labor at different education levels to analyze skill polarization; some studies have used job–task–skill data from official databases to analyze skill polarization; other studies measured skill levels through self-designed evaluation indicators and thus analyzed skill polarization [2,31,32,35].

Research on skill polarization and wage inequality found that the two phenomena are synchronized most of the time and are correlated [31,36]. Therefore, some scholars argued that the polarization of skill demand contributes to wage inequality to some extent [29], but other studies have found that wage gaps between different skill groups affect skill differentiation [35]. It can be seen that, although the relationship between skill polarization and occupational wages has been mentioned in existing studies [30–32], whether skill polarization leads to wage polarization still lacks sufficient empirical evidence.

To facilitate understanding and subsequent research, we define the concept of skill polarization in this study: after dividing occupational skills into socio-cognitive skills and sensory–physical skills, the occupational skills relied on by different occupations to perform the tasks of their job exhibit the phenomenon of clustering toward one of these categories. Based on the definition of skill polarization in this study, we argue that skill polarization is the external manifestation of changes in skill demand in different occupations. Career shifts are caused by changes in the demand for different types of labor [37]. This is because, to some extent, the acquisition of skills is a relatively slow process and therefore the supply of skills is also seen by researchers as remaining constant in the short run. However, the demand for skills in various occupations in the labor market is dominated by employers and it can be changed at any time to suit the employers' goals and pursuits. Therefore, we agree that skill upgrading and skill polarization are related to changes in skill demand rather than changes in skill supply [29].

Moreover, although there are many controversies and conflicts regarding the explanatory theories and impact mechanisms of job polarization, wage polarization and skill polarization in previous studies, scholars generally agree that the changes in the structure of labor force employment and earnings patterns experienced over the past decades have been largely driven by changes on the demand side of the labor market (employers) for skill levels and occupational expertise in different occupations, rather than by changes in the supply of skills on the supply side of the labor market (employees) [30]. Research on industry average wages has also found that aggregate demand associated with a particular industry has a strong positive effect on average earnings in that industry [38]. Therefore, the changes in the demand for skills in different occupations are reflected by the changes in the importance of skills in different occupations, thus showing the phenomenon of skill polarization. That means when the labor market exhibits skills polarization, that is, a preference for cognitive skills or physical skills in the skill importance of occupations, it

means that occupational demand for skills has become polarized. Under the condition that the skill supply of short-term labor remains unchanged, there will be a mismatch between the skill demand of occupations and the skill supply of the labor market. Skill mismatch is a significant source of inequality in earnings [39]. As the core skills required for an occupation to complete tasks change, this results in changes in the overall marginal output of the labor force for that occupation, affecting the labor demand of the occupation, resulting in changes in the labor supply and demand relationship of occupations (such as labor supply and demand imbalance). When the quantity of labor required by the skills of occupations is greater than the supply of labor, the wages of those occupations will rise and when the demand for an occupation's skills is less than the supply of labor, the wages will fall. It is confirmed that changes in skill utilization have an impact on wages [36]. Theoretically, this phenomenon of skill polarization based on changes in occupational demand for skills can affect the structure of labor employment and wage levels.

The construction of appropriate skill polarization proxies is the basis for the empirical research conducted in this study, which refers to the indicator design of [31,32], using the occupational socio-cognitive skills score and occupational effective use number of sensory–physical skills as the proxy indicator of the independent variable "skill polarization". The traditional method of measuring skill polarization focuses on classifying skill levels by the educational requirements or wage levels of occupations [31,35]. Although this method is convenient to operate, it essentially replaces changes in skills with changes in occupational wages. For example, suppose we use wage levels to reflect skill levels, at which point high-wage occupations are considered high-skill occupations. Thus, the traditional method of measuring skill polarization does not focus on changes in skills themselves. To overcome the problem that the coarse granularity of traditional indicators for measuring skill polarization may lead to ignoring important relationships between skills, this study adopts the indicator of occupational socio-cognitive skills scores to examine the degree of occupational skill differentiation from the perspective of social cognitive skills about the indicator design of [31,32]. Meanwhile, to avoid the indicator of [31] being unable to effectively classify occupations that rely on both socio-cognitive skills and sensory–physical skills, we use the indicator of the number of sensory–physical skills effectively used in occupations to examine the degree of differentiation of occupational skills from the perspective of sensory–physical skills.

Similar to [31,32,35], this study also focuses on the issue of skill polarization. However, unlike the former, we focus on the phenomenon of occupational skill polarization itself from the perspective of the skill needs of different occupations (i.e., differences in core skill requirements in different occupations), rather than using education and wages as criteria for classifying skill levels [35]. On the other hand, on the basis of [31] and [32], we extend the measures of skill polarization to answer whether occupational skill polarization is the cause of wage polarization. Therefore, combined with the existing literature, revealing that there is a significant correlation between skill polarization and occupational wages levels [31], this study proposes the following hypotheses:

**H1.** *The occupational socio-cognitive skills score can significantly and positively affect the wage level of the occupation; the higher the occupational socio-cognitive skills score, the higher the occupational wage level.*

**H2.** *The number of sensory–physical skills used effectively by an occupation has a significant and positive impact on the occupational wage level; the higher the number of sensory–physical skills used, the higher the occupational wage.*

**H3.** *Skill polarization can significantly and positively affect wage polarization in the labor market. The greater the skill polarization, the greater the wage polarization.*

The rest of the paper is organized as follows. Section 2 presents the data and methods of this study. Section 3 discusses our empirical results, including the regression results of the baseline model, the 2SLS regression results based on the instrumental variables (IV)

approach and the model robustness tests conducted by replacing the dependent variable. Finally, Section 4 provides the conclusion and discussion of this paper.

## 2. Materials and Methods

### 2.1. Data Sources

O*NET database. The O*NET database is a regularly updated annual database organized by the U.S. Department of Labor that provides the importance of abilities, knowledge, skills, work activities, work context and work values for each occupation in the Standard Occupational Classification (SOC) (the eight-digit O*NET-SOC occupation code) on a scale from 1 to 5, 1 indicating not important at all and 5 meaning extremely important. Referring to the design of [31,32], this paper uses the corresponding abilities, knowledge, skills and work activities for each occupation to describe occupational skills; therefore, each occupation has a corresponding 161 occupational skills. It should be pointed out that the skills mentioned subsequently in this paper are simplified representations of the 161 occupational skills defined here as "abilities, knowledge, skills and work activities" rather than the narrowly defined "skills" in the original O*NET database.

Based on the official classification of the O*NET database and combined with [32] classification criteria for socio-cognitive skills and sensory–physical skills, we divide the 161 skills into 63 sensory–physical skills and 98 socio-cognitive skills and denote the set of 98 socio-cognitive skills by $s \in SocioCog$. We use O*NET annual data from 2009 to 2021. The term $onet(o,s)$ denotes the importance of the skill $s \in S$ to the occupation $o \in O$. Additionally, referring to [31,32], we calculate the revealed comparative advantage (RCA) of each skill in the occupation $o$ to determine whether the corresponding skill is "effectively used" by the occupation, using the following formula:

$$rca(o,s) = \frac{onet(o,s) / \sum\limits_{s' \in S} onet(o,s')}{\sum\limits_{o' \in O} onet(o',s) / \sum\limits_{o' \in O, s' \in S} onet(o',s')}. \tag{1}$$

Occupational Employment and Wage Statistics (OEWS) database. The OEWS survey database is an annually updated database of the Bureau of Labor Statistics, which details the national average annual wage and hourly wage for each occupation for the previous year corresponding to the six-digit SOC occupation code (corresponding to the first six digits of the O*NET-SOC occupation code). Therefore, we use the OEWS survey data from 2009 to 2021 to match the O*NET database by using the six-digit SOC occupation codes. Since the O*NET database revised some occupation codes in 2010 and 2019 by adding, deleting, decomposing and merging some occupations, we further remove those occupations with missing values of the annual average wage in the OEWS database based on excluding the occupations that were completely added or that disappeared in that period. Finally, the panel data of 587 occupations over 13 consecutive years were obtained.

In addition, we downloaded the unemployment rate, direct investment abroad, export price index and labor force participation rate data from the U.S. Bureau of Labor Statistics website and the U.S. Census website for the period 2009 to 2021.

### 2.2. Variable Descriptions

#### 2.2.1. Wage Polarization

Over the years, there have been many studies on wage polarization. Some studies use the wages of occupations as a proxy indicator to reflect wage polarization [31,32,40] and others use the relative wages of different skill groups as a proxy for wage polarization [35]. In addition, some studies used different expressions such as "wage" and "average wage" in the analysis using wage data [2,11,21]. Considering that the study referenced in this paper used the occupational annual salary provided by the BLS to analyze wage polarization [31], this study refers to this data source and indicator design and we use the average annual

wages of occupations in the period 2009–2021 from the OEWS database provided by the BLS as the dependent variable.

### 2.2.2. Skill Polarization

Although many studies have explored the issue of skill polarization, including many attempts to use data to carry out empirical research, the existing studies have not yet reached a consensus on what indicators should be used to accurately describe skill polarization. According to the definition of skill polarization in this study, we draw on the classification method proposed by the skill-biased technological change (SBTC) hypothesis, defining socio-cognitive skills as high skills and sensory–physical skills as low skills. Correspondingly, occupations that mainly rely on socio-cognitive skills to complete work tasks are high-skilled occupations and those occupations that rely on sensory–physical skills are low-skilled. The demand for socio-cognitive skills and sensory–physical skills in different occupations increasingly shows a high reliance on one of them and the polarization of skills is demonstrated by the divergence of skills in the "socio-cognitive" and "sensory–physical" skill clusters in the labor market. Therefore, this study refers to the indicator design of [31,32] and uses the occupational socio-cognitive score, which reflects the level of demand for socio-cognitive skills in different occupations, as one of the proxies to evaluate the phenomenon of skill polarization in the labor market. The polarization of skills is distinguished from the perspective of the dependence of the occupation on socio-cognitive skills: the higher the socio-cognitive score of an occupation, the more it relies on socio-cognitive skills to perform tasks; on the contrary, the lower the socio-cognitive score of an occupation, the more it relies on sensory–physical skills to perform tasks. We calculate the socio-cognitive skills score for occupation $o$ by Equation (2).

$$cognitive_o = \frac{\sum\limits_{s \in SocioCog} onet(o,s)}{\sum\limits_{s \in S} onet(o,s)}.$$ (2)

Equation (2) treats the socio-cognitive skills score of an occupation as the ratio of the sum of the importance scores of all social cognitive skills in the occupation $o$ to the sum of the importance of all skills in it and $cognitive_o \in (0,1)$. However, the occupational socio-cognitive skills score only measures the polarized extent from the "one pole" of skill polarization, i.e., social cognitive skills. According to Equation (2), if an occupation has a high socio-cognitive skills score, it must mainly rely on social cognitive skills; however, the relatively low socio-cognitive skills score of an occupation may not only be due to a small numerator and a large denominator, but may also be because both the numerator and denominator are large, but the denominator has a larger value, meaning that an occupation relies on both social cognitive skills and sensory–physical skills. Subject to the strong hypothesis of the indicator, however, when an occupation has a low socio-cognitive skills score, the indicator cannot accurately determine whether the low score on the indicator is due to the occupation's primary reliance on sensory–physical skills or a reliance on both socio-cognitive and sensory–physical skills. A typical example is the profession of surgeon, which requires both high socio-cognitive skills as well as sensory–physical skills and should be classified as a high-skilled occupation. However, according to Equation (2), surgeons do not have a high socio-cognitive skills score. The authors of [31] were also aware of this problem, so they constructed the routine labor indicator from the perspective of the sensory–physical skills score of occupations, namely $\sum\limits_{s \in Routine} onet(o,s) / \sum\limits_{s \in S} onet(o,s)$, in an attempt to portray the phenomenon of skill polarization from both poles, respectively. However, the routine labor indicator is equal to $1 - cognitive_o$, so it still cannot fundamentally solve this problem.

To effectively overcome this problem, this study uses the number of sensory–physical skills used effectively in an occupation as a second proxy for skill polarization. Drawing on the design of [32], when the $rca(o,s) > 1$ of a sensory–physical skill in a certain

occupation is calculated according to Equation (1), this sensory–physical skill is considered to be effectively used in that occupation and thus the number of effectively used sensory–physical skills in the corresponding occupation is calculated. On the one hand, this is because the number of sensory–physical skills effectively used in occupations describes skill polarization from the other end of the skill dichotomy, i.e., the sensory–physical skills. Theoretically, the higher the number of sensory–physical skills used effectively in occupations with similar skill demands, the higher the occupational wage level. On the other hand, it can be seen from Equation (2) that the number of sensory–physical skills effectively used in an occupation can partially affect the size of the denominator and this is to some extent related to the occupational socio-cognitive skills score but does not uniquely determine its the level. Therefore, the phenomenon of skill polarization is evaluated jointly by two indicators, the occupational socio-cognitive skills score and the number of sensory–physical skills used effectively by occupations.

### 2.2.3. Control Variables

Unemployment rate. In the field of macroeconomics and microeconomics, many studies have confirmed a correlation between unemployment and wages [41,42]. The unemployment rate reflects the relationship between labor supply and demand in the labor market, which indirectly affects occupational wage levels. Generally, the higher the unemployment rate, the higher the number of workers competing for the same position, which leads to a fall in the wage level of the corresponding occupation. On the other hand, the unemployment rate also reflects the state of economic development. When the unemployment rate increases, it means that the overall economic development level is relatively poor and the wages that occupations can provide to employees will be relatively lower. Therefore, we use the unemployment rate as one of the control variables in this study.

Labor force participation rate. The relationship between labor force participation rate and wages is an important topic in the field of labor economics [43]. Similar to the unemployment rate, the labor force participation rate positively reflects the supply level of the labor market and also indirectly affects the wage levels of occupations. Therefore, we use the U.S. labor force participation rate in the corresponding year as a control variable to reflect the level of labor supply and demand.

Direct investment abroad. It is generally believed that direct investment abroad is related to the domestic wage level [44]. From the perspective of total investment, the share of OFDI correspondingly represents the loss of the share of domestic capital investment in the country. In general, when the amount of direct investment abroad increases, it implies a decrease in domestic capital investment, which will hurt the employment demand in the domestic labor market. Therefore, we use U.S. direct investment abroad in the corresponding year as a control variable reflecting the state of the economy.

Export price index. The EPI is a measure of exogenous price shocks in export trade industries and is a key factor affecting export trade, which can affect wages [24]. From the perspective of international trade, the export price index affects a country's share of foreign exports, which in turn indirectly affects the country's industrial structure and leads to changes in the structure of labor employment and occupational wages. In general, a higher export price index leads to a lower share of foreign export trade, which inhibits the development of industries engaged in export trade and affects labor force employment and wage levels in related industries. Therefore, we use the export price index of the corresponding year as a control variable reflecting the level of trade.

### 2.3. Empirical Methodologies

As elaborated in Chapter 2 of this paper, the research question of this study is whether skill polarization can cause wage polarization in the labor market. Therefore, a simple linear regression model of the following form is first considered.

$$W = \alpha + \beta S + \theta X + \varepsilon. \tag{3}$$

where $W$ represents wage polarization, $S$ reflects skill polarization, $X$ is control variables and $\varepsilon$ is the random error term.

Next, Equation (3) is further extended to the following formula based on the selection of indicators for each variable in this study.

$$ln(wage)_{ot} = \alpha_1 + \beta_1 socicogs_{ot} + \beta_2 epskill_{ot} + \theta_1 X_{ot-1} + \varepsilon_{ot}. \tag{4}$$

where $ln(wage)_{ot}$ represents the logarithm of the wage level of occupation $o$ in year $t$, $socicogs_{ot}$ represents the socio-cognitive skills score of occupation $o$ at $t$ and $epskill_{ot}$ represents the number of sensory–physical skills effectively used by occupation $o$ in period $t$. Additionally, $\beta_1$ and $\beta_2$ are the estimated coefficients that this study focuses on, which reflect the effect of skill polarization on wage polarization. $X_{ot-1}$ is the control variables in year $t-1$, including the unemployment rate ($unempr_{ot-1}$), labor force participation rate ($lfpr_{ot-1}$), direct investment abroad ($lgDIA_{ot-1}$) and export price index ($expi_{ot-1}$). It should be pointed out that skill polarization represents only a shift in occupational demand for skills and this change in occupational demand for skills only affects the employment and wages of various occupations in the short run due to changes in demand. After that, the continuous internal labor market supply and demand relationship and the overall economic environment affect occupational wages and employment in the long term. Therefore, the baseline model in this study focuses on the relationship between the current occupational socio-cognitive skills score, the number of sensory–physical skills used effectively in occupations and the phenomenon of wage polarization. In addition, the control variables selected in this study are macro-level variables that reflect the level of economic development and the supply and demand of labor. The impact of these variables on the labor market is usually considered to have some lagged effects; therefore, the control variables in the baseline model are selected with a first-period lag to prevent reverse causality. Additionally, $\varepsilon_{ot}$ also represents the random error term. The definitions and descriptions of all variables are given in Table 1.

**Table 1.** Variable definitions.

| Variable | Indicator | Indicator Symbol | Unit | Definition |
|---|---|---|---|---|
| **Dependent variable** | | | | |
| Wage polarization | Occupation wage(nominal) | lg(wage) | USD | Average annual salary per occupation from 2009 to 2021, logarithm. |
| **Independent variable** | | | | |
| Skill polarization | Socio-cognitive score of each occupation | socicogs | % | Based on the raw O*NET data, skills are classified as socio-cognitive or sensory–physical and then the socio-cognitive score of each occupation is calculated, which is within the range of (0,1). |
| | Number of sensory–physical skills effectively used per occupation | epskill | 1 | The revealed comparative advantage (RCA) of each skill in an occupation is calculated; the skill is used effectively if rca > 1. The sum of the number of senso-physical skills effectively used in each occupation is then taken. |
| **Control variable** | | | | |
| Unemployment rate | | unempr | % | U.S. annual unemployment rate from 2009 to 2021. |
| Direct investment abroad | | lg(DIA) | $ | Direct investment abroad of the United States from 2009 to 2021, logarithm. |
| Export price index | | expi | % | US export price index, 2009–2021. |
| Labor force participation rate | | lfpr | % | Labor force participation rate in the United States, 2009–2021. |

We also perform descriptive statistics on the data characteristics of the main variables and calculate the correlations between the variables. Table 2 shows the detailed results. The two proxy indicators of the independent variable skill polarization and the control variables selected in this study have significant correlations with the occupational wage level of the proxy of wage polarization; there is also a significant negative correlation between the two proxies of the independent variable skill polarization: the occupational socio-cognitive skills score and the number of sensory–physical skills effectively used in occupations. This suggests that the two proxies of the independent variable selected for this study can represent the "two ends" of skill polarization. Our data are short panels, so we use LLC and HT methods for testing homogeneous panel hypotheses and Fisher-type for testing heterogeneous panel hypotheses. The results of the unit root tests, including the LLC, HT and Fisher-type tests, for lg(wage), socicogs, epskill, unempr, lg(DIA), expi and lfpr are a stationary series themselves, almost all significant at the 1% level and reject the null hypothesis of the existence of unit root.

**Table 2.** Descriptive statistics and variable correlations.

| Variable | lg(wage) | Socicogs | Epskill | Unempr | lg(DIA) | Expi | Lfpr |
|---|---|---|---|---|---|---|---|
| Mean | 10.087 | 0.659 | 31.256 | 0.065 | 12.471 | 127.197 | 0.632 |
| Std.dev. | 0.466 | 0.0673 | 18.65 | 0.020 | 1.776 | 6.225 | 0.010 |
| Min | 9.811 | 0.527 | 0 | 0.0367 | 6.385 | 117.4 | 0.617 |
| Max | 12.65 | 0.791 | 61 | 0.0961 | 13.24 | 139.4 | 0.654 |
| Observations | 7631 | 7631 | 7631 | 7631 | 7631 | 7631 | 7631 |
| lg(wage) | 1 | | | | | | |
| socicogs | 0.487 *** | 1 | | | | | |
| epskill | −0.474 *** | −0.981 *** | 1 | | | | |
| unempr | −0.112 *** | −0.003 | −0.012 | 1 | | | |
| lg(DIA) | −0.043 *** | −0.001 | −0.005 | 0.424 *** | 1 | | |
| expi | 0.024 ** | 0.000 | 0.002 | −0.028 ** | 0.060 *** | 1 | |
| lfpr | −0.148 *** | −0.003 | −0.013 | 0.615 *** | 0.098 *** | −0.307 *** | 1 |
| LLC | −11.855 *** | $-3.1 \times 10^2$ *** | −65.005 *** | −26.563 *** | −61.095 *** | −9.815 *** | −33.512 *** |
| HT | 0.762 *** | 0.767 ** | 0.752 *** | 0.647 *** | 0.027 *** | 0.341 *** | 0.783 *** |
| Fisher Inverse Chi$^2$ | 2751.150 *** | 3757.785 *** | 2992.092 *** | 3655.213 *** | 3231.166 *** | 2625.449 *** | 2320.756 *** |
| Fisher Inverse normal | −25.207 *** | −31.637 *** | −28.664 *** | −41.218 *** | −36.918 *** | −30.127 *** | −26.337 *** |
| Fisher Inverse logit t | −25.898 *** | −39.662 *** | −30.494 *** | −40.988 *** | −35.890 *** | −28.368 *** | −24.418 *** |
| Fisher Modified inv. Chi$^2$ | 32.548 *** | 53.322 *** | 37.520 *** | 51.205 *** | 42.454 *** | 29.954 *** | 23.666 *** |

Note: ** and *** denote statistical significance at the 5% and 1% level, respectively.

## 3. Results and Discussion

To investigate the effect of skill polarization on wage polarization, this chapter is divided into three parts: first, the effect of skill polarization on wage polarization is studied based on a fixed-effect model; second, the endogeneity problem of the regression model is verified; and third, model robustness tests are conducted by substituting dependent variables.

### 3.1. Baseline Results

After checking the panel data, the data in this study are found to constitute a balanced panel. Since the lagged effect of the dependent variable itself is not considered in the model, the normative static panel analysis method is adopted. First, we choose the regression method by comparing the mixed regression model, fixed-effects model and random-effects model and the detailed results are shown in Table 3. This shows that the results of the F-test strongly reject the hypothesis of mixed regression being acceptable and the LSDV result shows that each dummy variable is significant (*p*-value is 0.0000). Hence, there are considered to be individual effects. Likewise, the result of the LM test suggests that a random-effects model should be chosen. Therefore, the Hausman test is further performed to determine the final regression model and the results are shown in Table 4. The *p*-value is 0.0000, indicating that a fixed-effects model should be used.

**Table 3.** The selection of mixed regression, fixed-effect and random-effect models.

| | OLS | FE | RE |
|---|---|---|---|
| **Variable** | **lg(wage)** | **lg(wage)** | **lg(wage)** |
| socicogs | 3.596 *** | 0.478 *** | 1.081 *** |
| | (3.36) | (3.22) | (4.46) |
| epskill | 0.001 | 0.002 *** | 0.001 ** |
| | (0.21) | (4.70) | (2.21) |
| L. unempr | −0.485 *** | −0.493 *** | −0.491 *** |
| | (−11.25) | (−12.27) | (−13.90) |
| L.lg(DIA) | −0.008 *** | −0.008 *** | −0.008 *** |
| | (−32.75) | (−22.36) | (−35.73) |
| L. expi | −0.004 *** | −0.004 *** | −0.004 *** |
| | (−32.76) | (−34.06) | (−35.74) |
| L.lfpr | −5.669 *** | −5.709 *** | −5.703 *** |
| | (−44.58) | (−71.73) | (−62.79) |
| Constant | 12.626 *** | 14.675 *** | 14.283 *** |
| | (14.34) | (121.01) | (75.20) |
| F test | | F = 795.42 | |
| | | (*p* = 0.0000) | |
| LM test | | | Chibar$^2$ = 37,464.37 |
| | | | (*p* = 0.0000) |
| *N* | 7044 | 7044 | 7044 |
| $R^2$ | 0.259 | 0.684 | 0.683 |

Note: *t* statistics in parentheses. ** and *** denote statistical significance at the 5% and 1% level, respectively.

**Table 4.** The results of the Hausman test.

| | FE | RE |
|---|---|---|
| **Variable** | **lg(wage)** | **lg(wage)** |
| socicogs | 0.478 *** | 1.081 *** |
| | (3.22) | (7.65) |
| epskill | 0.002 *** | 0.001 *** |
| | (4.70) | (3.92) |
| L. unempr | −0.493 *** | −0.491 *** |
| | (−12.27) | (−12.11) |
| L.lg(DIA) | −0.008 *** | −0.008 *** |
| | (−22.36) | (−22.13) |
| L. expi | −0.004 *** | −0.004 *** |
| | (−34.06) | (−33.72) |
| L.lfpr | −5.709 *** | −5.703 *** |
| | (−71.73) | (−70.91) |
| Constant | 14.675 *** | 14.283 *** |
| | (121.01) | (119.73) |
| Hausman test | Chi$^2$ = 148.98 | |
| | (*p* = 0.0000) | |
| *N* | 7044 | 7044 |
| $R^2$ | 0.684 | |

Note: *t* statistics in parentheses. *** denote statistical significance at the 1% level.

Based on Equation (4) of the baseline model, we also constructed model (2) by considering the first-period lag of the occupational socio-cognitive skills score and of the number of sensory–physical skills used effectively for occupations as new independent variables, respectively, and substituting the second-period lag of the independent and control variables into the model to construct model (3), thus forming the following equations:

$$ln(wage)_{ot} = \alpha_2 + \beta_3 socicogs_{ot-1} + \beta_4 epskill_{ot-1} + \theta_2 X_{ot-1} + \varepsilon_{ot}.$$
$$ln(wage)_{ot} = \alpha_3 + \beta_5 socicogs_{ot-2} + \beta_6 epskill_{ot-2} + \theta_3 X_{ot-2} + \varepsilon_{ot}. \tag{5}$$

Table 5 reports the regression results of the fixed-effects model of the impact of skill polarization on wage polarization. From the results, both the occupational socio-cognitive skills score and its first-period lagged effect on the wage level of the occupation are significantly positive at the 10% level, with coefficients of 0.478 and 0.479, respectively, while the coefficients of the lagged second-period are not significant. This indicates that in the baseline model, the higher the occupational socio-cognitive skills score, the higher the corresponding occupation wage level. The coefficient of the occupational socio-cognitive skills score is 0.478, indicating that the occupational wage level increases by 4.78% for every 0.1 unit increase in the occupational socio-cognitive skills score after adding the relevant control variables. This shows that the polarization of skills plays an important role in the process of wage polarization and hypothesis H1 is validated. This result supports the positive correlation between skill polarization and wage level as reflected by the occupational socio-cognitive skills score proposed by [31]. On the one hand, from the perspective of social skills and cognitive skills, occupations that require high levels of social and cognitive skills generate a higher skill premium, driving up occupational wages [45,46]. On the other hand, a possible mechanism for this result is that skill polarization is an external manifestation of the divergence in skills demand for different occupations. Since the adjustment of skill supply in the labor market requires a certain period; it is generally believed that the skill supply of labor in the short term is constant, which leads to a mismatch between skill demand and supply in the labor market, resulting in an imbalance between labor supply and demand. When the demand for labor is greater than the supply in the labor market, wages will rise; when the amount of labor demand is less than the labor supply for a given occupation, the wage of the occupation will fall.

**Table 5.** Baseline model results.

| | Baseline Model | Model (2) | Model (3) |
|---|---|---|---|
| Variable | lg(wage) | lg(wage) | lg(wage) |
| socicogs | 0.478 * (1.84) | | |
| L. socicogs | | 0.479 * (1.93) | |
| L2.socicogs | | | 0.227 (1.05) |
| epskill | 0.002 *** (2.66) | | |
| L. epskill | | 0.002 *** (2.92) | |
| L2.epskill | | | 0.001 * (1.77) |
| L. unempr | −0.493 *** (−13.94) | −0.488 *** (−13.69) | |
| L2.unempr | | | −4.422 *** (−47.66) |
| L.lg(DIA) | −0.008 *** (−35.83) | −0.008 *** (−35.96) | |
| L2.lg(DIA) | | | −0.001 ** (−2.77) |
| L. expi | −0.004 *** (−35.89) | −0.004 *** (−35.85) | |
| L2.expi | | | 0.002 *** (15.45) |
| L.lfpr | −5.709 *** (−63.24) | −5.708 *** (−63.32) | |
| L2.lfpr | | | 3.242 *** (17.14) |
| Occupation FE | Yes | Yes | Yes |
| Constant | 14.675 *** (74.14) | 14.670 *** (77.32) | 8.644 *** (42.92) |
| $N$ | 7044 | 7044 | 6457 |
| $R^2$ | 0.684 | 0.684 | 0.796 |

Note: $t$ statistics in parentheses. *, ** and *** denote statistical significance at the 10%, 5% and 1% level.

In all three models, the coefficient of the effect of the number of sensory–physical skills used effectively in occupations, another proxy indicator of skill polarization, on the occupational wage level is significantly positive, being 0.002, 0.002 and 0.001, respectively. Additionally, the coefficients for the current period and the lagged first-period are significant at the 1% level. In the baseline model, for example, the wage level of the occupation increases by 0.2% for each increase in the number of sensory–physical skills effectively used by the occupation, indicating that hypothesis H2 passes validation. This result supports the finding by [31] that an increase in the routine labor index, which is essentially equal to $1 - cognitive_o$, increases the wage levels of occupations when considering both the occupational social cognitive skills score and the routine labor index. This phenomenon can be explained by the following mechanism: a higher number of skills that occupations can effectively use indicates that the occupation requires more types of skills for labor, requires more challenging tasks to be performed and accordingly requires higher wages. The occupations with a large number of effective uses of sensory–physical skills are mainly concentrated in occupations that rely on sensory–physical skills to complete work tasks. Additionally, as the number of sensory–physical skills that these occupations require effectively increases, wages increase. However, it should be noted that, although the influence coefficients of the occupational socio-cognitive skills score and the number of sensory–physical skills used effectively for occupations in terms of the occupational wages are both significantly positive, the effect of the occupational socio-cognitive skills score on the occupational wage level (0.478) is much larger than that of the number of sensory–physical skills used effectively in occupations (0.002). This finding corroborates Autor's (2015) conclusion that wage growth in high-skilled occupations is greater than that in low-skilled occupations and provides new evidence for Mazzolari and Ragusa's (2013) suggestion that the increased demand for low-skilled service sector jobs by high-wage workers is responsible for job polarization [1]. This result may be because wage increases for high-skilled workers and the rising demand for outsourced household production activities have an impact on the low-skilled labor market through "consumption spillovers", which exert upward pressure on the wages of the low-skilled workers who primarily provide these services, driving up their wages [1,21].

When both the independent and control variables take the second-period lag, the effect of the occupational socio-cognitive skills score on occupational wages is insignificant, while the significance of the effect of the number of sensory–physical skills effectively used by occupations on wages drops sharply, being only significant at the 10% level. This change supports our theoretical hypothesis on the relationship between skill polarization and wages, that is, in the long run, the supply and demand relationship in the labor market is a dynamic process affected by multiple factors and shifts in skills demand in different jobs are represented by skill polarization. This will only affect labor market supply and demand in the short run, which will ultimately affect wages across occupations. Hypotheses H1 and H2 are both validated and the increase in the occupational socio-cognitive skills score (representing high skills) and the number of sensory–physical skills used effectively by occupations (representing low skills) are both associated with increases in occupational wages. Therefore, the greater the degree of skill polarization, the higher the corresponding occupational wage level, that is, skill polarization has a positive effect on wage polarization and hypothesis H3 is also verified.

Both the first-period and second-period lags of the control variables can significantly affect the occupational wage level. In the baseline model, the coefficient of the influence of the unemployment rate of the previous year (one lag period) on occupational wages is significantly negative at the 1% level, which means that an increase in the unemployment rate will reduce the occupational wage level overall. An increase in the labor force participation rate has a significant negative effect on domestic wages. For every 1% increase in direct investment abroad, wages in domestic occupations fall by 0.8%. The export price index increases by 1%, reducing occupational wages by 0.4%. These results confirm that factors such as the level of economic development at the macroeconomic level, the level of

export and import trade, the scale of investment abroad and the relationship between labor supply and demand in the labor market have been considered in the existing literature and it has been found that they can consistently affect occupational wages and job polarization.

### 3.2. Endogeneity Issues

Next, the endogeneity issue of the model will be considered. If skill polarization can significantly affect wage polarization, then, intuitively, it is reasonable to believe that the objective reality of wage polarization causes changes in the demand and supply of occupational skills, which in turn have an inverse effect on skill polarization. Therefore, the occupational socio-cognitive skills score as a proxy for skill polarization may not be strictly exogenous. To solve such a problem, we introduce the lag term of the occupational socio-cognitive skills score (considering both the first-period and second-period lags) as instrumental variables (IV) in the regression model. The parameters are estimated using the 2SLS method and the results are shown in Table 6. According to the first column in Table 6, the regression results of the first stage of the 2SLS method are shown and the following results can be obtained: first, the LM statistic is 565.329, which is significant at the 1% significance level, rejecting the original hypothesis that the IVs are not identifiable. Second, the *p*-value of the Cragg–Donald Wald F statistic is much less than 1%, rejecting the null hypothesis of weak IVs. Third, the results based on the Anderson–Rubin Wald test further verified the correlation between the IVs selected by the model and the occupational socio-cognitive skills score. Fourth, the *p*-value of the Hansen J statistic was 0.8672, allowing us to accept the null hypothesis that IVs are exogenous. The coefficients of IVs (first-period and second-period lags of the occupational socio-cognitive skills score) are 0.660 and −0.330, respectively, and both are significant at the 1% level, indicating that the lagged one-period occupational socio-cognitive skills score has a promotional effect on the baseline period, while lagged two-period occupational social cognitive skills scores have a negative effect.

**Table 6.** Instrumental variable 2SLS regression.

| | Stage 1 | Stage 2 |
| --- | --- | --- |
| **Variable** | **socicogs** | **lg(wage)** |
| L.socicogs | 0.660 ***<br>(30.40) | |
| L2.socicogs | −0.330 ***<br>(−3.72) | |
| socicogs | | 0.527 **<br>(2.23) |
| epskill | −0.001 ***<br>(−12.91) | 0.002 ***<br>(4.14) |
| L. unempr | −0.003<br>(−1.56) | −0.491 ***<br>(−8.86) |
| L. lg(DIA) | −0.000<br>(−1.00) | −0.008 ***<br>(−26.19) |
| L. expi | 0.000<br>(0.00) | −0.004 ***<br>(−32.77) |
| L. lfpr | −0.013 ***<br>(−2.75) | −5.695 ***<br>(−47.00) |
| Constant | | 14.629 ***<br>(82.30) |
| LM statistic | | 565.329 *** |
| Cragg–Donald Wald F statistic | | 2932.721 *** |
| Anderson–Rubin Wald test | | 2.47 * |
| Hansen J statistic | | 0.028 (*p* = 0.8672) |
| Occupation FE | | Yes |
| Observation | | 6457 |
| $R^2$ | | 0.650 |

Note: *t* statistics in parentheses. *, ** and *** denote statistical significance at the 10%, 5% and 1% level.

The regression results of the second stage of the 2SLS method are presented in Table 6, indicating that the occupational socio-cognitive skills score has a significant positive effect on wage polarization at the 0.5% significance level with an impact coefficient of 0.527, which is consistent with the results of the baseline model; therefore, the endogeneity problem of the model is effectively mitigated and the robustness of the model results is enhanced.

### 3.3. Robustness Check

Regarding model robustness tests, common practices include substituting proxy indicators for dependent variables, replacing proxies with independent variables and so on. In the previous sections of the regression analysis, we used the logarithm of occupational wages without a price deflator (namely nominal wages) as the proxy of the dependent variable. Considering the measurement error and the robustness of the regression results, we construct the robustness check model (1) by adding the employment number of occupations to the basic model, use the logarithm of the real wages of each occupation calculated according to the level of the dollar in 2009 to replace the dependent variable in the base model for constructing the robustness check model (2) and construct the robustness check model (3) based on the robustness check model (1) and (2). The relevant results are detailed in Table 7.

**Table 7.** Robustness tests.

| Variable | Robustness Test Model 1 | Robustness Test Model 2 | Robustness Test Model 3 |
|---|---|---|---|
| | lg(wage) | lg(rwage) | lg(rwage) |
| socicogs | 0.495 ** (1.98) | 0.380 * (1.66) | 0.392 * (1.76) |
| lg(employ) | −0.078 ** (−7.83) | | −0.0517 *** (−5.31) |
| epskill | 0.002 ** (2.52) | 0.001 ** (2.53) | 0.001 ** (2.43) |
| L. unempr | −0.576 *** (−14.81) | −0.473 *** (−13.39) | −0.528 *** (−13.66) |
| L.lg(DIA) | −0.008 *** (−35.97) | −0.001 ** (−5.76) | −0.001 *** (−5.79) |
| L. expi | −0.004 *** (−35.37) | −0.002 *** (−18.30) | −0.002 *** (−17.98) |
| L.lfpr | −5.591 *** (−63.34) | −0.389 *** (−4.31) | −0.311 *** (−3.59) |
| Occupation FE | YES | YES | YES |
| Constant | 15.418 *** (70.50) | 10.941 *** (61.69) | 11.433 *** (57.07) |
| Observation | 7044 | 7044 | 7044 |
| $R^2$ | 0.701 | 0.192 | 0.223 |

Note: $t$ statistics in parentheses. Occupation employment takes the logarithm. *, ** and *** denote statistical significance at the 10%, 5% and 1% level.

The results in Table 7 show that the occupational socio-cognitive skills score and the number of sensory–physical skills used effectively by occupations have a significant positive correlation with wage polarization and the results are consistent with those of the baseline model. This indicates that the analytical results of this study are robust.

## 4. Conclusions

Skill polarization is the external manifestation of changes in occupational demand for skills. Changes in the demand side of the labor market for different skill levels and professional expertise in different occupations cause changes in the labor employment structure [30]. However, in the existing literature, skill polarization is studied as a concomitant phenomenon of job polarization and there is not enough empirical research to support whether skill polarization leads to wage polarization. This study constructs the occupa-

tional socio-cognitive skills score and an RCA index of occupational skills based on the measurement method proposed by [31] using the occupation–skill data from the American O*NET database, constructs proxies to measure skill polarization using O*NET data from 2009 to 2021 and matches occupational codes based on the occupation–employment data from the same period. We explore the impact of the relationship between skill polarization and wage polarization by analyzing panel data for the first time and test the reliability and explanatory power of the indicator of the occupational socio-cognitive skills score proposed in [31].

Different from [21], which examined the phenomenon of wage polarization in the U.S. from a macro perspective of globalization, this paper focuses on the aspect of skill polarization, which has long been mentioned but lacks relevant empirical research. We find that, as poles of skill polarization, the increase in the occupational socio-cognitive skills score and the number of sensory–physical skills used effectively by occupations are associated with a rise in occupational wages. Empirical results suggest that skill polarization has a positive effect on wage polarization. This finding provides new evidence for the positive correlation between skills polarization and employment polarization proposed by Alabdulkareem et al. (2018), which uses a wider range of data to test the reliability of the occupational socio-cognitive skills scores in measuring the high-skill side of skill polarization. New influences are provided to understand the phenomenon of wage polarization exhibited in the U.S. labor market as found by [20]. In contrast to [31,32], the indicator of the number of sensory–physical skills effectively used by occupations designed in this paper not only evaluates occupations that rely on only sensory–physical skills or socio-cognitive skills, but also identifies those occupations that rely on both socio-cognitive skills and sensory–physical skills. In addition, the positive effect of the socio-cognitive skill score of occupation on wages is higher than the number of perceived physical skills effectively used by the occupation, providing new insights into the idea that increased demand for low-skilled service industry jobs by high-wage workers may lead to polarization [1].

This is because the labor market exhibiting skills polarization means that the occupational demand for skills has become polarized, as skill mismatch is an important source of earnings inequality [39]. When the skill demand for occupations polarizes, there is a certain gap between the skill supply and demand of occupations, resulting in some skills having a higher supply than demand (e.g., welding skills in electronics production), while other skills have a lower supply than demand (such as skills for developing AI programs), since socio-cognitive skills are generally considered to be acquired through education and require a certain amount of time to learn. Therefore, socio-cognitive skills exhibit higher demand than supply, which also leads to higher wages in occupations that depend on socio-cognitive skills. At the same time, the increase in demand for occupations that rely on socio-cognitive skills increases the demand for low-skilled occupations that serve high-skilled ones, which raises the need for occupations that depend on sensory–physical skills, thereby increasing the wages of those occupations.

We also find that an increase in the unemployment rate can reduce the wage level of occupations; an increase in the labor force participation rate has a significant negative impact on occupational wages; direct investment abroad and the export price index can inhibit the wage level of occupations. We also test our findings using the IV method and the substitution of dependent variables to ensure the robustness of the regression results.

At the same time, our findings supplement the influencing factors of the wage polarization phenomenon and enrich the relevant literature on empirical research on wage polarization. The existing literature on wage polarization either analyzes it from a macroeconomic perspective, such as technology, trade and the offshoring of work tasks, or at an organizational level (such as corporate-level restructuring), but there is currently no research on wage polarization from the perspective of occupational skills supply and demand. This study fills this gap in the existing literature, enriches the analysis of factors affecting the phenomenon of wage polarization and introduces internal factors of the labor

market into the study of wage polarization, which provides a new perspective and theory for understanding the phenomenon of wage polarization. Moreover, this paper also shows that the factors affecting wage polarization in the labor market are complex and researchers and policymakers should not only focus on macroeconomic factors but also pay attention to the impact of internal factors in the labor market on the employment structure.

In addition, there are some limitations of this study that need to be discussed in subsequent work. First, the sample in this study spans from 2009 to 2021 and the data are taken from the national level in the United States. In future research, we can try to use longer-term data to verify the causal effects and further use data from the state or Metropolitan statistical area level to consider the regional heterogeneity of the effects of skill polarization. Second, the selection of control variables is limited by the difficulty of data collection. This study currently considers only four labor-market-related factors, such as the unemployment rate and export price index. Factors directly related to occupational wages, including offshoring data and significant policy implications, have not been taken into account. For instance, it is commonly believed that economic performance and policies have an impact on domestic labor markets and our study has not considered the impact of the 2008 financial crisis and the U.S. government's "manufacturing repatriation" policy, which can be further discussed in future studies. Finally, in terms of research methodology, structural equation modeling can be further constructed to verify and communicate the results of this study in the future.

**Author Contributions:** Conceptualization, H.J. and Q.G.; methodology, H.J.; software, H.J.; validation, Q.G.; investigation, H.J.; resources, Q.G.; writing–original draft preparation, H.J.; writing–review and editing, Q.G.; supervision, Q.G. All authors have read and agreed to the published version of the manuscript.

**Funding:** This research received no external funding.

**Institutional Review Board Statement:** Not applicable.

**Informed Consent Statement:** Not applicable.

**Data Availability Statement:** The data presented in this study are available in https://www.onetonline.org/ (accessed on 3 June 2022) and https://www.bls.gov/oes/ (accessed on 3 June 2020).

**Conflicts of Interest:** The authors declare no conflict of interest.

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
