# Peer review of "Does Skill Polarization Affect Wage Polarization? U.S. Evidence 2009–2021"

_sustainability, doi:10.3390/su142113947_

Round 1

Reviewer 1 Report

The paper attempts to examine “The relationship between skill polarization and wage polarization in the case of U.S”. After reviewing, I find that this paper is very interesting. The paper run numerous techniques, especially 2SLS. For better contribution to the literature, I have some revisions that are good for enhancing the quality of the manuscript.

·         The Abstract is well written; however, I suggest to authors recommend some policy implication at the end of the Abstract.

·         The introduction is also well written, authors developed a sound background of the study with research hypothesis.

·         The research methods section is well written, however control variables are lack of justification.

·         The results of the study are interesting; I suggest to authors compare it with previous studies.

·         In the conclusion section, authors should mention the limitation/future research path.

Author Response

General comments: The paper attempts to examine “The relationship between skill polarization and wage polarization in the case of U.S”. After reviewing, I find that this paper is very interesting. The paper run numerous techniques, especially 2SLS. For better contribution to the literature, I have some revisions that are good for enhancing the quality of the manuscript.

1.Comment: The Abstract is well written; however, I suggest to authors recommend some policy implication at the end of the Abstract.

1.Reply: We gratefully appreciate your valuable suggestion, and we have adopted this recommendation and added the policy recommendations to the Abstract.

The content is modified as follows (see P1):

Our findings confirm that skills polarization has a positive effect on wage polarization, providing new insights into understanding employment inequality in the labor market. Authorities should focus more attention on increasing the earnings of the low- and middle-skilled workers, especially through vocational skills training to increase the number of sensory-physical skills that can ultimately mitigate wage polarization.

2.Comment: The introduction is also well written, authors developed a sound background of the study with research hypothesis.

2. Reply: Thank you so much for your careful check.

3. Comment: The research methods section is well written, however control variables are lack of justification.

3. Reply: Thank you for your rigorous comment. We agree that the choice of control variables is a current limitation of this study. In conjunction with this research question, from the perspective of sound model design, the control variables in this study need to consider not only labor market-related variables but also the impact that factors related to the forces of globalization may have on the skill demand and wages of occupations; therefore, we have attempted to collect data on offshoring in the United States for the last decade or so. Unfortunately, however, limited by the difficulty of data collection, we have not collected direct data on U.S. participation in the global economy, such as production outsourcing data, between 2009 and 2021 for the time being. Therefore, only relevant variables reflecting U.S. economic engagement with other countries, such as OFDI and export price index, can be selected. In the related studies, both OFDI and export price index reflect the economic interactions between the U.S. and other economies to some extent, and both indicators are related to production outsourcing, which can reflect the influence of globalization factors to some extent. Second, among the variables related to occupational wages, we currently select two variables related to occupational wages, the unemployment rate, and the labor force participation rate. Some studies have revealed that both the unemployment rate and labor force participation rate affect the wage level, so we believe that the unemployment rate and labor force participation rate are usually related to the employment situation in the labor market, reflecting some extent the employment situation in the labor market and the domestic economic situation, and the wages of each occupation are relatively higher when the economic situation is relatively good. Finally, we add to the article a description of the literature and rationale for the chosen control variables for the research question.

The content is modified as follows (see P6, P15-16):

In the field of macroeconomics and microeconomics, many studies have confirmed a correlation between unemployment and wages [41, 42].

The relationship between labor force participation rate and wages is one of the important topics in the field of labor economics [43].

It is generally believed that direct investment abroad is related to the domestic wage level [44].

The EPI is a measure of exogenous price shocks in export trade industries and is a key factor affecting export trade, which can affect wages [24].

Second, the selection of control variables is limited by the difficulty of data collection. This study currently considers only four labor market-related factors such as the unemployment rate and export price index. Factors directly related to occupational wages, including offshoring data and significant policy implications, have not been taken into account.

4. Comment: The results of the study are interesting; I suggest to authors compare it with previous studies.

4. Reply: Thank you for your nice suggestion. We adopt this suggestion and add a comparison with other relevant literature in the conclusion section. This paper is systematically summarized by comparing it with existing literature in terms of research perspective, indicator design, and research findings.

The relevant modifications are as follows (see P14-15):

Different from [21], which examined the phenomenon of wage polarization in the U.S. from a macro perspective of globalization, this paper focuses on the aspect of skill polarization that has been long-mentioned but lacks relevant empirical research. We find that, as poles of skill polarization, the increase in the occupational sociocognitive skills score and the occupational effective use number of sensory-physical skills are associated with occupational wages rise. Empirical results suggest that skill polarization has a positive effect on wage polarization. This finding provides new evidence for the positive correlation between skills polarization and employment polarization proposed by Alabdulkareem et al. (2018), which uses a wider range of data to test the reliability of the occupational sociocognitive skills scores in measuring the high-skilled side of skill polarization. New influences are provided to understand the phenomenon of wage polarization exhibited in the U.S. labor market as found by [20]. In contrast to [31,32], the indicator of the number of sensory-physical skills effectively used by occupations designed in this paper not only evaluates occupations that rely on only sensory-physical skills or sociocognitive skills but also identifies those occupations that rely on both sociocognitive skills and sensory-physical skills. In addition, the positive effect of the sociocognitive skill score of occupation on wages is higher than the number of perceived physical skills effectively used by the occupation, providing new insights into the idea that increased demand for low-skilled service industry jobs by high-wage workers may lead to polarization [1].

5. Comment: In the conclusion section, authors should mention the limitation/future research path.

5. Reply: We gratefully appreciate your valuable comment. We adopt this suggestion and expand the conclusion section regarding the limitations of the study as well as future research.

The relevant changes are as follows (see P15-16):

First, the sample in this study spans from 2009 to 2021 and the data is at the national level in the United States. In future research, we can try to use longer-term data to verify the causal effects, and further use state or Metropolitan statistical area level data to consider the regional heterogeneity of the effects of skill polarization. Second, the selection of control variables is limited by the difficulty of data collection. This study currently considers only four labor market-related factors such as the unemployment rate and export price index. Factors directly related to occupational wages, including offshoring data and significant policy implications, have not been taken into account. For instance, it is commonly believed that economic performance and policies have an impact on domestic labor markets, and our study has not considered the impact of the 2008 financial crisis and the U.S. government’s “manufacturing repatriation” policy, which can be further discussed in future studies. Finally, in terms of research methodology, structural equation modeling can be further constructed to verify and communicate the results of this study in the future.

Reviewer 2 Report

The paper deals with an important issue but there are several major flaws with the paper, which are discussed below.

-Contributions. The general contribution of the paper to the literature is not very strong, since the link between skill polarization and wage polarization has already received much attention in the literature. If the paper's particular contribution is to develop and apply an alternative measure of skill polarization in an occupational context, why is the proposed measure better than existing measures? A good justification is needed if this is the real contribution of the paper. Furthermore, it is unclear what the value added is of using this alternative measure. For example, does it help explain more of the observed wage polarization than alternatives?

-Measurement. Average annual wages of occupations is used as a proxy for wage polarization. This is a key variable in the analysis, yet there is no proper justification for using this variable or robustness checks to compare the results using alternative measures that differently reflect polarization in wages.

-Model setup. There is insufficient critical discussion and justification for the empirical model. An example on p.8 is that the dependent variable is not considered to be persistent over time. Yet wages and wage polarization are well known to be persistent. A related issue is that the estimations might generate spurious correlations because the stationarity assumption might not hold in the context of highly persistent wages. These are important issues given the empirical nature of the paper.

-Presentation of results. Some of the estimation results seem rather inconsistent (e.g. some of the t-statistics differ in Tables 3-4 for the same RE estimations). It is unclear what exactly is presented in the tables (t-statistics or standard errors?). There are frequent errors in presentation around the tables and in the notes. These are basic errors, and academic rigour is lacking in this respect.

-Economic significance. Another issue is the lack of meaningful discussion about the economic significance of the results. How large and important are the main estimates economically in driving wage polarization? We learn on p.9 that a 1 unit increase in occupational sociocognitive skills generates a 47.8% increase in occupational wage, but the change in skills required to generate such polarization is very large relative to its standard deviation. Therefore, current reporting of results is not that useful in relation to detailing their economic significance. Moreover, we do not know how much of the variation in wages is explained by the main skill polarization variables because results are not reported excluding fixed effects or including a statistic like the partial-R2 for example.

-Endogeneity issues. Discussion of endogeneity issues should be more central within the paper. Discussion of relevant endogeneity issues, selection of endogenous variables in the equation, choice of instruments, etc, should be discussed in more detail earlier on. There are numerous variables that are reputed to be important in this nexus which are not sufficiently controlled for, especially in relation to globalization forces.  

Author Response

General comments: The paper deals with an important issue but there are several major flaws with the paper, which are discussed below.

1. Comment: Contributions. The general contribution of the paper to the literature is not very strong, since the link between skill polarization and wage polarization has already received much attention in the literature. If the paper's particular contribution is to develop and apply an alternative measure of skill polarization in an occupational context, why is the proposed measure better than existing measures? A good justification is needed if this is the real contribution of the paper. Furthermore, it is unclear what the value added is of using this alternative measure. For example, does it help explain more of the observed wage polarization than alternatives?

1.  Reply: Thank you for your rigorous comment. We consider this issue to be critical. First, we have complied with the problem of this article by adding a justification of the innovation and benefits of adopting the new measurement method in the Introductory section. Second, in Section 2.2, the original article discusses the literature origin, design ideas, and benefits of the new method. Finally, it should be added that although there have been many studies in the existing literature that focus on the link between skill polarization and wage polarization, the traditional approach to measuring skill polarization has mainly been to analyze skill polarization about employment polarization, and wage polarization by dividing it by the occupational demand for educational attainment; or itself by dividing skill levels by wage levels. However, this coarse-grained distinction may ignore the important relationship between skills that affect employees' adaptability. Therefore, Alabdulkareem et al. (2018) returned to the labor market demand for occupational skills itself, measured skill polarization in terms of both social cognitive skills and sensory physical skills, and analyzed it by linking it to the average degree requirements of the occupation and found that the new measure of skill polarization could well explain the differences in the demand for degrees across occupations, with social cognitive skills, indicate higher education requirements for different occupations, while occupations with more lenient degree requirements tend to rely on sensory physical skills. However, since the sociocognitive skills score and sensory-physical skills score indicators designed by Alabdulkareem et al. (2018) are essentially two calculation perspectives of the same indicator, i.e., the sum of the two indicators is equal to 1. This leads to the inability of this indicator design to effectively classify occupations that rely on both social cognitive skills and physical perceptual skills. This paper uses the indicator of the number of sensory-physical skills effectively used in occupations to examine the degree of differentiation of occupational skills from the perspective of sensory physical skills. Also, referring to the design of Alabdulkareem et al. (2018), the indicator of sociocognitive skills scores of occupations is used to examine the degree of differentiation of occupational skills from the perspective of social cognitive skills.

The modifications to the original paper are as follows (in P3 and P5).

The traditional method of measuring skill polarization focuses on classifying skill levels by the educational requirements of occupations or the wage levels of occupations [31,35]. Although this method is convenient to operate, it essentially replaces changes in skills with changes in occupational wages. For example, suppose we use wage levels to reflect skill levels, at which point high-wage occupations are considered high-skill occupations. Thus, the traditional method of measuring skill polarization does not focus on changes in skills themselves. To overcome the problem that the coarse granularity of traditional indicators for measuring skill polarization may lead to ignoring important relationships between skills, this study adopts the indicator of occupational sociocognitive skills scores to examine the degree of occupational skill differentiation from the perspective of social cognitive skills about the indicator design of [31,32]. Meanwhile, to avoid that the indicator of [31] cannot effectively classify occupations that rely on both sociocognitive skills and sensory-physical skills, we use the indicator of the number of sensory-physical skills effectively used in occupations to examine the degree of differentiation of occupational skills from the perspective of sensory-physical skills.

A typical example is the profession of surgeon, which requires both high sociocognitive skills as well as sensory-physical skills and should be classified as a high-skilled occupation. While according to equation (2), surgeons do not have a high sociocognitive skills score.

2. Comment: Measurement. Average annual wages of occupations is used as a proxy for wage polarization. This is a key variable in the analysis, yet there is no proper justification for using this variable or robustness checks to compare the results using alternative measures that differently reflect polarization in wages.

2. Reply: Thank you so much for your careful check. We thank the expert for pointing out the problem regarding the indicator of average annual wage. First, we have added a literature citation and justification in Section 2.2. It should be further noted that in existing studies using occupation-level wage data, there are indeed different expressions such as "wage" and "average wage". However, both the annual wage and the hourly wage of occupation are essentially the average wage, i.e., the average hourly wage or the average annual wage. For example, in studies such as Autor and Dorn (2013) and Frey and Osborne (2016), which used the average annual wage of occupation for data analysis, these articles used the expression "wage" in their theoretical interpretation on the one hand and returned to the expression "average wage" in their data analysis on the other hand. Second, the OEWS database of the BLS is a classic source of data used by many studies, such as Autor and Dorn (2013) and Frey and Osborne (2016), which use the BLS occupational wage data in their studies. Whereas only data related to average annual wages and average hourly wages are provided in the OEWS database, studies conducted using this database are essentially average wages regardless of the presentation. Finally, in Section 3.3, we rerun the fixed effects regression by replacing the dependent variable of the model and find that the study results are robust, which indicates that the average annual wage used in the base model is reliable as the dependent variable.

The modifications to the original paper are as follows (in P5):

In addition, some studies used different expressions such as "wage" and "average wage" in the analysis using wage data [2, 11, 21]. Considering that the study referenced in this paper used the occupational annual salary provided by the BLS to analyze wage polarization [31], this study refers to this data source and indicator design, and we use the average annual wages of occupations in the period 2009-2021 in the OEWS data provided by the BLS as the dependent variable.

3. Comment: Model setup. There is insufficient critical discussion and justification for the empirical model. An example on p.8 is that the dependent variable is not considered to be persistent over time. Yet wages and wage polarization are well known to be persistent. A related issue is that the estimations might generate spurious correlations because the stationarity assumption might not hold in the context of highly persistent wages. These are important issues given the empirical nature of the paper.

3. Reply: We gratefully appreciate your valuable suggestion. Regarding the issue of possible spurious correlations due to unsteady panel data, we also considered this problem in our study and did a stability test of panel data, but it was not reported in the original paper. In response, we added the results of the correlation stability test to Table 2 and explained them.

The modifications are as follows (see P8 for details):

The results of the unit root tests, including the tests of LLC, HT, and Fisher-type, for lg(wage) are a stationary series themselves which are significant at the 1% level.

4.  Comment: Presentation of results. Some of the estimation results seem rather inconsistent (e.g. some of the t-statistics differ in Tables 3-4 for the same RE estimations). It is unclear what exactly is presented in the tables (t-statistics or standard errors?). There are frequent errors in presentation around the tables and in the notes. These are basic errors, and academic rigour is lacking in this respect.

4. Reply:  We feel sorry for the inconvenience brought to the reviewer. We accept this recommendation. It should be explained that the content in brackets in Table 3 to Table 7 should be t-statistics, but due to an oversight, the comments section of the original Table 3 to Table 7 incorrectly labeled the content in brackets as t-statistics while labeling it as standard errors, and there was also a problem with sentence repetition. We have checked all the contents of the table and made corrections.

The relevant modifications resulted in the following (see P9-11,13-14 for detailed table locations).

5. Comment: Economic significance. Another issue is the lack of meaningful discussion about the economic significance of the results. How large and important are the main estimates economically in driving wage polarization? We learn on p.9 that a 1 unit increase in occupational sociocognitive skills generates a 47.8% increase in occupational wage, but the change in skills required to generate such polarization is very large relative to its standard deviation. Therefore, current reporting of results is not that useful in relation to detailing their economic significance. Moreover, we do not know how much of the variation in wages is explained by the main skill polarization variables because results are not reported excluding fixed effects or including a statistic like the partial-R2 for example.

5. Reply: Thank you for your nice suggestion. We fully agree with the experts' comments. First, we have used a more appropriate presentation in the interpretation of the base model regression results in Section 3.1, i.e., for every 0.1 unit increase in occupational social cognitive skill score, the wage level of the occupation increases by 4.78%. Second, the fixed effects regression results in the original article were all controlled for individual fixed effects, but they were not presented in the relevant tables, and we now add individual fixed effects content displayed in Table5.

The modifications are as follows (in P10-11, P14):

The coefficient of the occupational sociocognitive skills score is 0.478, indicating that the occupational wage level increases by 4.78% for every 0.1 unit increase in the occupational sociocognitive skills score after adding the relevant control variables.

6. Comment: Endogeneity issues. Discussion of endogeneity issues should be more central within the paper. Discussion of relevant endogeneity issues, selection of endogenous variables in the equation, choice of instruments, etc, should be discussed in more detail earlier on. There are numerous variables that are reputed to be important in this nexus which are not sufficiently controlled for, especially in relation to globalization forces.

6. Reply: We understand the reviewer’s concern. In this study, we have tried to overcome the problem of endogeneity as much as possible. First, the problem of endogeneity has been controlled for in this paper in two ways: one is through the IV approach by introducing the occupational sociocognitive skills score lag terms (considering both lag one and lag two) as IVs into the model and estimating the parameters by using the 2SLS method, and the other is through the substitution of the dependent variable. The results of these two methods show that the result of this study is robust. Second, we acknowledge that the choice of control variables is a limitation of the present study. Combined with the fact that this study is concerned with the issue of the relationship between skill polarization and wage polarization, the control variables in this study need to consider the possible effects of factors related to the forces of globalization on the skill demand and wages of occupations in addition to those related to the labor market, from the perspective of sound model design. Unfortunately, at present, we have only selected two variables related to occupational wages, unemployment rate and labor force participation rate. We believe that the unemployment rate and labor force participation rate are usually related to the employment situation in the labor market, and to some extent, they reflect the employment situation in the labor market and the domestic economic situation, and the wages of each occupation will be relatively higher when the economic situation is relatively good. Meanwhile, limited by the difficulty of data collection, we temporarily do not collect direct data on U.S. participation in the global economy between 2009 and 2021, such as offshoring data, so we can only choose variables that reflect the correlation between U.S. economic interactions with other countries, for example, direct investment abroad and export price index. Both OFDI and export price indexes reflect some extent the economic engagement between the U.S. and other economies, and both of them are related to production outsourcing.

Reviewer 3 Report

Formal evaluation of the work:

1. The article consists of 16 pages, including the actual text of the work (pages 1-15) and references (pages 15-16). The article has been divided into 4 main parts: 1. Introduction; 2. Materials and Methods; 3. Results and Discussion, 4. Conclusions. Subchapters 2 and 3 are further divided into subsections. According to the reviewer, the layout of the work used is correct.

2. In the work authors uses regression models for panel data, the 2SLS method and the robustness check of the models. According to the reviewer, the analyzes were performed correctly.

3. Reference contains 42 items directly related to the topic of the work and, in the opinion of the reviewer, it is a sufficient number.

4. The work contain seven tables that have been prepared correctly.

Substantive evaluation of the work:

Research article of great scientific value. The title of the work corresponds to its content. The aim of the work and research hypotheses have been clearly presented in subsection 1. (Introduction). In the opinion of the reviewer, they were confirmed in the presented research results. The research methodology is also described in detail (section 2). The method of conducting the research and the depth of the analyzes carried out deserve praise. In the opinion of the reviewer, the article is suitable for publication.

Author Response

General comments: Substantive evaluation of the work: Research article of great scientific value. The title of the work corresponds to its content. The aim of the work and research hypotheses have been clearly presented in subsection 1. (Introduction). In the opinion of the reviewer, they were confirmed in the presented research results. The research methodology is also described in detail (section 2). The method of conducting the research and the depth of the analyzes carried out deserve praise. In the opinion of the reviewer, the article is suitable for publication.

1. Comment: The article consists of 16 pages, including the actual text of the work (pages 1-15) and references (pages 15-16). The article has been divided into 4 main parts: 1. Introduction; 2. Materials and Methods; 3. Results and Discussion, 4. Conclusions. Subchapters 2 and 3 are further divided into subsections. According to the reviewer, the layout of the work used is correct.

1. Reply: Thank you for your rigorous consideration.

2. Comment: In the work authors uses regression models for panel data, the 2SLS method and the robustness check of the models. According to the reviewer, the analyzes were performed correctly.

2. Reply: Thank you for your rigorous comment.

3. Comment: Reference contains 42 items directly related to the topic of the work and, in the opinion of the reviewer, it is a sufficient number.

3. Reply: Thank you so much for your careful check.

4. Comment: The work contain seven tables that have been prepared correctly.

4. Reply: Thank you for your rigorous consideration.

Round 2

Reviewer 2 Report

Brief comments on the revised version:

-The endogeneity treatment could be better justified. It is unclear why only one regressor is considered endogenous and not others. Also, it is unclear what your 2SLS specification is. For example, have fixed effects been omitted? If so, we might be concerned about bias arising from model misspecification.

-A whole raft of different stationarity tests are employed but it is unclear why all tests are relevant / useful in your sample. Also, it is unclear what is actually being tested. For example, what are the assumptions made about deterministic trends when conducting these tests for stochastic non-stationarity?

-Another issue here is why only the dependent variable is considered important for determining stationarity in the regression model, since non-stationarity can arise from variables on both sides of the equation.  
